# Role of Intracellular Amyloid β as Pathway Modulator, Biomarker, and Therapy Target

**DOI:** 10.3390/ijms23094656

**Published:** 2022-04-22

**Authors:** Lucia Gallego Villarejo, Lisa Bachmann, David Marks, Maite Brachthäuser, Alexander Geidies, Thorsten Müller

**Affiliations:** 1Department of Molecular Biochemistry, Cell Signalling, Ruhr University Bochum, 44801 Bochum, Germany; lucia.gallegovillarejo@ruhr-uni-bochum.de (L.G.V.); lisa.bachmann@ruhr-uni-bochum.de (L.B.); david.marks@ruhr-uni-bochum.de (D.M.); maite.brachthaeuser@ruhr-uni-bochum.de (M.B.); alexander.geidies@ruhr-uni-bochum.de (A.G.); 2Institute of Psychiatric Phenomics and Genomics (IPPG), LMU University Hospital, LMU Munich, 80336 Munich, Germany

**Keywords:** Alzheimer’s disease, intracellular amyloid β, APP, amyloid, neurodegeneration

## Abstract

The β- and γ-secretase-driven cleavage of the amyloid precursor protein (APP) gives rise to the amyloid β peptide, which is believed to be the main driver of neurodegeneration in Alzheimer’s disease (AD). As it is prominently detectable in extracellular plaques in post-mortem AD brain samples, research in recent decades focused on the pathological role of extracellular amyloid β aggregation, widely neglecting the potential meaning of very early generation of amyloid β inside the cell. In the last few years, the importance of intracellular amyloid β (iAβ) as a strong player in neurodegeneration has been indicated by a rising number of studies. In this review, iAβ is highlighted as a crucial APP cleavage fragment, able to manipulate intracellular pathways and foster neurodegeneration. We demonstrate its relevance as a pathological marker and shed light on initial studies aiming to modulate iAβ through pharmacological treatment, which has been shown to have beneficial effects on cognitive properties in animal models. Finally, we display the relevance of viral infections on iAβ generation and point out future directions urgently needed to manifest the potential relevance of iAβ in Alzheimer’s disease.

## 1. Introduction

Alzheimer’s disease (AD) is a neurodegenerative disease that was first described by the psychiatrist Alois Alzheimer in 1906 [1]. Today, it is the most common cause of dementia and is clinically characterized by a progressive impairment of behavioral and cognitive functions including memory, comprehension, language, attention, reasoning, and judgment [2]. A central hallmark of the disease is the presence of amyloidogenic plaques, formed by deposition, accumulation, and aggregation of the amyloid β peptide (Aβ) in the brain. The second well-described pathophysiological characteristic are neurofibrillary tangles, which result from hyperphosphorylation of the microtubule-associated Tau protein, consequently leading to cytoskeletal changes in neurons [3]. Both events display a progressive development, typically beginning in the entorhinal cortex in the hippocampus and primarily affecting the medial temporal lobe and associative neocortical structures, whose impairment causes the aforementioned symptoms due to neuronal cell death [4].

According to the amyloid hypothesis, amyloidogenic plaque formation is caused by an imbalance in the production and accumulation of the Aβ peptide resulting from the processing of the amyloid precursor protein (APP) [5]. In non-pathological conditions, the transmembrane protein APP is predominantly cleaved by an α-secretase, thereby producing fragments of different sizes, which are described to be non-cytotoxic [6]. In case APP is processed within the amyloidogenic pathway, sequential cleavage by β- and γ-secretase causes the formation of 38–42 aa long amyloid peptide species believed to be secreted into the extracellular space. These soluble monomeric forms are known to aggregate into soluble oligomers, which then form insoluble fibrils that are finally deposited as amyloidogenic plaques [7,8,9]. Although amyloid plaques were studied as the main pathological event in AD, multiple lines of evidence (such as the accumulation of oligomeric species in animal AD models disrupting synaptic plasticity and causing memory impairment) have pointed to prefibrillar Aβ oligomer species playing a central role in AD pathogenesis. Due to its potential relevance, the Aβ oligomer hypothesis and its possible implications have been largely studied over the last several years [10].

More recently, the pathophysiological role of intracellular Aβ (iAβ) has aroused increasing interest, as iAβ was found to affect intracellular pathways, pointing to a potential relevance of iAβ accumulation for very early molecular aberrations that precede even amyloid plaque and neurofibrillary tangle formation. Within this review, we highlight iAβ as an intracellular modulator and pathological biomarker, illustrate iAβ genesis and drugs affecting it, and address viral infections and their impact on iAβ generation.

## 2. Mechanisms of Intracellular Aβ Generation and Enrichment

While the mere presence of accumulated iAβ is accepted as an early occurring phenomenon in AD preceding the extracellular accumulation of plaque-forming Aβ, less is known about the mechanisms predominantly leading to the excessive accumulation within cells, especially in neurons [11,12,13,14,15]. Accumulated iAβ can be considered as the result of an imbalance between intracellular production of Aβ, the import of extracellular Aβ, and the clearance of Aβ by diverse mechanisms. However, it remains elusive which pathways play the dominant role in leading to excessive and toxic aggregation [16,17,18,19,20,21,22,23,24]. Accordingly, a number of pathways have been studied with regard to their contribution to the internalization of Aβ, thereby focusing on Aβ accumulation, reuptake, and clearance as the causes for reaching toxic levels of iAβ.

New insights revealed that long iAβ isoforms, such as Aβ_45_, are remaining and aggregating within neurons, while in contrast, shorter Aβ_42_ isoforms are secreted, resulting in the surge of intracellular aggregation with time due to age-related changes in Aβ processing [25]. Indeed, it was demonstrated that iAβ contains long Aβ_45_ peptides that accumulate in mitochondria, endosomes, and autophagosomes, which increase with age and upon glutamate treatment. It was speculated that age-related signaling for inhibition of C-terminal trimming by γ-secretase increases Aβ_45_ levels whose hydrophobic properties promote aggregation [25].

A closer look at the endocytic processing of Aβ suggests a mechanism of internal accumulation that is not issued from the uptake of Aβ peptides but from the dysfunction of metalloproteases of the endothelin-converting enzyme (ECE) family [26]. These enzymes reside within exosomes and normally limit the accumulation of iAβ. Through the pharmacological inhibition of these metalloproteases, Aβ and iAβ levels were enriched in human neuroblastoma SH-SY5Y cell lines, primary neurons, and organotypic brain slices from an AD mouse model. Furthermore, iAβ oligomer formation was promoted in a process independent of the internalization of secreted Aβ [26]. These findings suggest a crucial role for multivesicular bodies, from which exosomes derive, as intracellular sites of Aβ degradation by these enzymes. Thus, it was proposed that ECE dysfunction could lead to the accumulation of intraneuronal Aβ aggregates and their subsequent release into the extracellular space via exosomes [26].

Further, there is strong evidence that the reuptake of Aβ occurs primarily via the endocytic pathways. Accordingly, it was demonstrated that soluble Aβ_40_ and Aβ_42_ primarily use endocytosis as the major, possibly even the only, pathway of entry [27]. A strong correlation of the peptides with lysosomes was observed, and Aβ_42_ uptake was two times more efficient than Aβ_1–40_ uptake. However, both peptides have been shown to be largely using the same uptake paths, which are predominantly clathrin- and dynamin-independent but actin-dependent [27].

Specifically, in neurons, FcγRIIb2, a variant of Fcγ-receptor IIb (FcγRIIb), holds a critical function for the neuronal uptake of pathogenic Aβ, as demonstrated in an AD mouse model [28]. In more detail, it was postulated that the Fc(y)RIIb2-mediated oligomeric Aβ_42_ uptake is involved in the di-leucine-dependent receptor-mediated endocytosis, which resulted in the accumulation of excess oligomeric Aβ_42_ mainly in the lysosome. This uptake was attenuated by TOM1, a FcRIIb2-binding protein that represses the receptor recycling, which additionally enhances the hypothesis that FcγRIIb is responsible for iAβ neuronal uptake and suggests a potential molecular mechanism underlying iAβ accumulation [28].

Finally, the accumulation of iAβ is associated with the glymphatic system, also known as the paravascular pathway, primarily playing a pivotal role in the clearance of a major fraction of extracellular Aβ, which is mediated by astroglia aquaporin 4 (AQP4) [29]. Based on studies in various mouse models, including brain aging, AD, and mild traumatic brain injury [30,31,32], which found that glymphatic clearance malfunction is related to mislocalization of AQP4 caused by reactive astrogliosis, the glymphatic clearance ability among AQP4^−/−^/APP/PS1 mice, APP/PS1 mice, AQP4^−/−^ mice, and wild-type (WT) mice was compared. Moreover, the effect of selective elimination of microglial cells or downregulation of apolipoprotein E (apoE) expression on the Aβ burden in the frontal cortex was investigated. As a result, it was possible to demonstrate that AQP4 deletion exacerbates glymphatic clearance impairment, accompanied by an increase in accumulation of iAβ and apoE in the APP/PS1 brain, and the knockdown of apoE reduces iAβ levels in APP/PS1 mice with or without AQP4 [29].

## 3. Intracellular Aβ Clearing by the Ubiquitin–Proteasome System

Ubiquitination, especially the ubiquitin–proteasome system (UPS), which plays a pivotal role in the prevention of iAβ deposition and the resulting influence on the cellular burden, is directed by the ubiquitin E3 ligase enzymes [33,34]. While there are other mechanisms involved in the regulation of these proteins, the ubiquitination pathway is a central process that involves the selection of key lysine residues of target protein E3 ligases for ubiquitin attachment. The lysine components of proteins can act as a docking site for the ubiquitin attachment, and depending on the type of poly-ubiquitin chains, the cellular fate of the target protein is determined [35]. The Aβ production is influenced by amyloid β precursor protein (AβPP) ubiquitination and proteasomal degradation of the β- and γ- secretases. The structure of AβPP was modeled, and its topology was determined to investigate the impact of lysine residues on AβPP stability and the resulting influence on disease disposition [36]. It was found that K351 is the best target for AβPP ubiquitination, and also conserved amino acids were detected, which play a crucial role in AβPP ubiquitination [36]. Furthermore, potential ubiquitination enzymes E1s, E2s, E3s, and deubiquitinating enzymes (DUBs) were identified, as well as their contribution to the complex interplay in the AβPP ubiquitination process, illuminating the AβPP ubiquitination, as well as the clearance of iAβ and possible targets for therapeutic approaches in AD [36,37,38,39].

## 4. Intracellular Aβ Affects Pathways Involved in Cellular Stress, Plasticity, and Receptor Function

For decades, research on Alzheimer’s disease focused on the well-known late-stage hallmarks, when cognitive symptoms arise. However, due to a continuing lack of effective therapeutic treatment, the research focus has shifted to the early stages of AD in order to understand the causative mechanisms of pathophysiological dysregulation and find potential pharmacological targets. In this regard, the accumulation of iAβ has become a highly relevant topic after different studies showed that it accumulates within neurons in the very early stages of the disease, e.g., in transgenic mouse models and humans [16,40]. Investigations in the last 5 years reported toxic effects on multiple cellular pathways, including calcium and synaptic dysregulation, inhibition of the ubiquitin–proteasome system, mitochondrial dysfunction, and activation of proinflammatory responses [41,42,43]. However, the underlying mechanism for the pathological effect of iAβ remains unclear.

Recent studies tested multiple approaches to assess calcium dysregulation as one of the most commonly described mechanisms causing synaptic dysfunction and cell death (Figure 1). Indeed, it was shown that iAβ_42_ aggregate formation is caused by the internalization of extracellular Aβ after interaction with membrane receptors or membrane permeation pathways [44]. Therefore, they designed a model to expose artificial membranes, isolated mitochondria, and neuronal cells to Aβ_42_ toxic forms and mutant (non-aggregating) Aβ_42_ peptides. The authors demonstrated that these non-aggregating isoforms were able to inhibit wild-type Aβ_42_ extracellular internalization, intracellular aggregation, and neuronal toxicity [44]. Thereby, they firstly confirmed the already described ability of soluble Aβ_42_ to interact with plasma membrane phospholipids, creating Ca^2+^ pores and thereby enhancing a cytotoxic calcium influx [45]. A significant reduction in the Aβ_42_ toxic capacity was shown using non-aggregating Aβ_42_ isoforms, pointing to a possible therapeutic target for early AD-related events. Using the same approach, the authors also studied another described iAβ_42_ effect, which is based on the oligomer–membrane interaction [16,46] that presumably leads to iAβ_42_ internalization in mitochondria [47], interaction with apoptotic proteins [48], reduction in the activity of respiratory enzymes such as cytochrome c oxidase (COX) [49], and ultimately, mitochondrial damage and cell death [50,51]. Mutant iAβ_42_ was able to reduce the capability of iAβ_42_ to interact with the mitochondrial membrane, impairing the iAβ_42_-mediated mitochondrial potential depolarization, decrease in COX activity, and subsequent cell toxicity. In accordance with these results, a combination of transcriptomic, proteomic, and metabolomic experiments indicated an intensive activation in oxytosis/ferroptosis non-apoptotic cell death pathways, which was caused by an increase in intracellular lipid peroxides and reactive oxygen species (ROS) in a high-level iAβ model [52]. Endoplasmic reticulum oxidative stress response and/or mitochondrial damage were proposed to be the main mechanisms contributing to the activation of oxytosis/ferroptosis in the MC65 nerve cells [52].

Based on previous results in transgenic mice models for iAβ accumulation [53,54], it was hypothesized that iAβ-induced BK channel suppression provokes a broadening of the action potentials enhancing Ca^2+^ influx, leading to a dysregulation in Ca^2+^ homeostasis and subsequent cell death (Figure 1). In their follow-up study, anti-APP or anti-Aβ-oligomer antibodies were found to be sufficient to reverse BK channel suppression, pointing to synergistic activity of both molecules, and driving attention to APP as a toxic agent. Interestingly, in order to develop a complementary approach for the investigation of the iAβ effect in Ca^2+^ homeostasis, Minicucci et al. designed a mathematical model [55]. They validated previous results of an interaction between iAβ and PLC triggering the production of IP_3_, which forces an abnormal release of Ca^2+^ from the ER after hyperactivation of IP_3_ receptors [56] (Figure 1). These studies on calcium dysregulation suggested a relevant effect of iAβ in neuronal excitability.

## 5. Impact of Intracellular Aβ on the Neuronal Architecture

Fernandez-Perez et al. specifically studied the impact of Aβ oligomers in the nucleus accumbens of double (APP/PS1) transgenic mice showing an increased iAβ accumulation. The nucleus accumbens is a brain area very closely related to the main affected regions in AD (such as the hippocampus and cortex) that plays a critical role in reward, cognition, learning, and emotional behavior [57,58,59]. Electrophysiological recordings in dissociated neurons and brain slices showed an association between iAβ and an increase in excitability at resting conditions, or predisposition to firing, which was in agreement with previous findings [60]. Additionally, a decrease in glycine receptors and postsynaptic markers was observed that correlated with a decrease in miniature synaptic currents and gly-evoked potentials together with a decrease in AMPA-evoked potentials. These results point to an impairment in both inhibitory and excitatory synaptic transmission. Additional results of the same group using AD brain-derived and synthetic Aβ oligomers after intracellular supply on hippocampal neurons revealed an increase in synaptic transmission, excitability, and neuronal synchronization. Interestingly, iAβ seemed to affect circuit levels, causing a “functional spreading” of hyperexcitability. In addition, a post-synaptic potentiation in AMPA currents by PKC-dependent mechanisms was reported. By analysis of synaptic failure in a rat model, researchers found a reduction in memory-function-related gene expression (Arc, c-fos, Egr1, and Bdfn) together with a blockage in CRTC1 translocation in the hippocampus [61]. These results were in accordance with the synaptic plasticity alteration by long-term potentiation (LTP) inhibition in the CA1 region of the hippocampus and in the neocortex caused by iAβ accumulation [62]. Notably, behavioral tests performed for visual discrimination, associative learning, and behavioral control showed severe impairment in learning visual–reward association. Further experiments confirmed the observed long-term memory and social behavior alteration in this model, which was concomitant with an increase in transcript levels of synaptic plasticity and memory genes such as Grin2b, Dlg4, Camk2b, and Syn1, emphasizing the role of iAβ in pre-plaque stages of the disease.

Comparably, Ochiishi et al. studied the effect of iAβ from cellular to behavioral level in a mice model designed to express a fusion Aβ-GFP protein without secretion signal that therefore accumulates inside the neurons [63]. Notably, iAβ caused increased Tau phosphorylation, altered spine morphology, and attenuated long-term-potentiation (LTP), without plaque formation, atrophy, or neuronal cell death, compared with control mice. Moreover, iAβ-carrying mice showed strong memory impairment at a young age but no alterations in locomotor activity, which is a common side effect observed in other AD mice models. Their results reinforce the aforementioned iAβ toxicity against synaptic function and subsequent memory impairment at a young age.

In addition to these findings, a role for iAβ_42_ in the disruption of axonal transport and, therefore, dysregulation of neuronal synaptic transmission was shown involving the mitogen-activated protein kinase (MAPK) [64]. In a biochemical characterization of iAβ accumulation on MAPK and morphogenetic signaling, the authors were able to show that increased iAβ_42_ expression leads to a significant reduction in ERK 1/2 phosphorylation and increased bone morphogenetic protein-2-dependent Smad 1/5/8 phosphorylation. Rescuing the iAβ_42_-mediated attenuation of MAPK signaling was possible with the small molecule PLX4032, which is a downstream enhancer of the MAPK pathway.

## 6. Intracellular Aβ as Driver of Neuroinflammation

The role of the immune system and inflammation in early AD pathology has become another intensively studied factor based on the evidence of inflammatory markers increasing in patients months before the occurrence of extracellular plaques (Figure 1). In addition, it was observed that Aβ_42_ treatment raises the inflammatory S100A9 signaling [65], which further relegates the extracellular plaque formation to a less relevant position. It is, therefore, possible to consider iAβ accumulation, as well as inflammatory signaling molecules such as S100A9 or IL-6, as causative for neuroinflammation in AD [65,66]. Indeed, upregulation of key inflammatory mediators (including IL-6, CCL2, CCL3, and CSF-1) at transcript and protein levels in Aβ-burdened (using extracts from pre-plaque APP transgenic rats) hippocampal neurons, compared with those extracted from WT animals, was found in AD transgenic animal models before plaque deposition [67]. The study also showed a correlation between neuronal expression of chemotactic and proinflammatory molecules and recruitment of activated microglia, highlighting the role of neurons as primary inflammatory agents in the early stages of the disease. These results were in accordance with the previously described increased production of pro-inflammatory cytokines (such as TNF-a and IFN-g) in CA1 neurons Aβ-burdened from a transgenic rat model for early pre-plaque AD conditions [68]. Correspondingly, studies focusing on the role of microglia demonstrated the preferential engagement of microglia with amyloid-burdened neurons in the early stages of AD [69]. Using a pre-plaque mice model, it was observed that the response of microglia to early amyloid accumulation translated into an increase in soma–soma, process–soma, and process–neurite interactions, preferably occurring in iAβ-positive neurons. Additionally, in microglia, phagocytic internalization on dendrites and axons was observed. Interestingly, TREM2 was not found to be required for these early structural and functional changes in microglia–neuron interactions, opposite to what is described for later stages of the disease, when microglia’s main function is described to be the clearance of amyloid species [70]. Similarly, an increase in necrotic markers has been noted in patients in the early stages of AD development, further considering early occurring necrotic signaling proteins such as phosphorylated MARCKS, activated by the HMGB1–TLR4 complex [71,72] as an additional pathological mechanism.

## 7. Intracellular Aβ as a Spreading Pathological Marker

Recent studies demonstrated that iAβ is present in elevated amounts in cells of patients affected by AD as one of the earliest pathologic markers preceding extracellular amyloid formation [17,71,73,74]. In addition, a natural accumulation of iAβ in the brain was shown to be age-dependent, indicating a higher risk to develop AD [75]. The relevance of iAβ as a pathological marker was further strengthened by the finding that the peptide was associated with cognitive impairment in familial AD (FAD) cases harboring a PSEN2 mutation [73,76]. Further, a connection of iAβ (and not plaque pathology) to cognitive impairment has been proven (and reversed followed by passive immunization) using anti-Aβ antibodies and genetic reduction in APP expression, reducing iAβ levels but not affecting plaque pathology in the brain of an animal model [77,78,79,80]. Notably, the reduction in iAβ levels reduced synaptic deficits and cognitive impairments in mouse models. These results yielded new studies regarding iAβ conglomerates and their spread. Interestingly, it was demonstrated that brain extracts from AD mouse models are able to induce iAβ conglomeration in 2D cell culture, subsequently inducing iAβ conglomeration in other cells for at least 10 passages [73]. This suggests that iAβ possesses infectious prion-like potency and induces a gain of function since iAβ is able to shift the balance toward amyloidogenic instead of non-amyloidogenic cleavage, which was shown by further iAβ accumulation [73,81,82]. Despite the prion-like potency, no prion protein typical proteinase K resistance was present, rather an increased resistance against denaturing substances such as SDS [73]. In summary, iAβ represents a potential new pathologic marker for AD. As it precedes amyloid plaque formation, it allows earlier detection of the disease and enables the follow-up of new disease-related findings and hypotheses.

## 8. Intracellular Aβ and Amyloidogenic Peptides in Viral Infections

Epidemiological studies indicated a link between lytic or latent viral infection and an increased risk for amyloid-associated diseases, e.g., AD or diabetes mellitus (DM) [83,84,85,86]. Interestingly, in some cases, this risk was reduced upon antiviral therapy [87]. Several viruses discussed in this context were also reported to specifically infect cell types with relevance for the pathology of amyloid diseases. Herpes zoster virus (HZV), for instance, infects different cell types that are also affected in AD, such as neurons and glia, and DM such as pancreatic cells, indicating the possible contribution to the observed disease phenotypes [88,89].

Apart from that, this connection also raises the question of whether these virally induced diseases are amyloid diseases themselves. Varicella-zoster virus (VZV) infection of quiescent primary human spinal astrocytes (qHA-sps) induced the production of intracellular amyloidogenic proteins (intracellular amylin, APP, and/or iAβ, and amyloid, which includes amyloid-like fibrillar structures, prefibrillar oligomers, and fibrils). Additionally, conditioned media of infected astrocytes promoted the process of aggregation in uninfected cells. An elevation of amyloidogenic protein levels was similarly reported in the plasma of acute zoster patients and for the CSF from VZV vasculopathy patients, compared with respective controls [90,91,92]. In addition to that, a significant enrichment of pathways associated with amyloid-associated diseases was shown upon VZV infection of human brain vascular adventitial fibroblasts (HBVAFs) [93]. The increased presence of amyloidogenic peptides such as Aβ and iAβ in infected cells further emphasizes the hypothesis that VZV vasculopathy is an amyloid disease.

Moreover, herpes simplex virus type 1 (HSV-1) viral DNA preferentially colocalized with Aβ plaques in ADs patients’ brains [83], and promotion of extracellular Aβ and iAβ accumulation, as well as Tau hyperphosphorylation, was observed in a variety of in vitro models [86,94,95,96,97,98,99,100]. Accumulation of amyloid could, in part, create a cytotoxic and proinflammatory environment, which could further affect neighboring cells by inducing a potentiation of chronic inflammation as observed in most patients.

Intriguingly, mouse model experiments revealed a close link between viral infection, the observed pathological changes, and cognitive impairment [101]. Therefore, besides other influencing factors, viral infection with VZV or HSV-1 increased the toxic amyloid burden and contributed to amyloid-associated disease progression. Even though amyloidogenic proteins were not detected extracellularly, increased intracellular amyloid could be released to the extracellular space upon apoptosis, thus “seeding” extracellular aggregation [16].

Contradictory to these results is a recent study that reported the absence of any significant correlation between amyloidogenic protein expression, as well as Aβ deposition, in the human brain of herpes simplex encephalitis patients. Additionally, a lack of specificity of AD-related pathological hallmarks for HSV-1-infected cell types or areas was demonstrated [102]. Consequently, the authors state that the vulnerability of the human CNS to virally induced neuropathology could be overestimated and suggest further analysis of this effect in more patients.

All in all, there have been numerous studies demonstrating that viruses can indeed promote the formation and accumulation of pathological features of amyloid diseases such as iAβ, and therefore, viral infections could be potential risk factors for the development of AD, DM, etc. The molecular basis of viral infection needs to be further investigated, as it could be essential for the identification of potential therapeutic targets for amyloid diseases. Nevertheless, recent opposing findings indicate that further investigation of the role of viral infection in amyloid diseases is crucial for uncovering the molecular mechanism. In order to properly address the question, human specimens particularly need to be the focus of these examinations.

## 9. Intracellular Aβ as a New Target for Therapeutic Drug Treatment

The lack of full understanding of the complex pathology of AD has so far impeded the development of efficient therapy and treatment options. Due to the seemingly complex nature of AD, it is difficult to find a target for therapy and pharmaceutical agents, as there are so many pathologic events coming along with AD. In addition, therapeutic drugs need to be selected and developed to minimize side effects and limit toxicity while still being able to pass the blood–brain barrier, which has led many promising candidates to fail on the way to clinical applications [103,104,105,106,107]. Since the accumulation of iAβ occurs during the very early stages of the disease, it might constitute a high-potential target for therapeutic drugs, and inhibition or clearance of iAβ at early stages might prevent irreversible damage to the neuronal cells in the brain, which could consequently slow down neurodegeneration. This potential has been recognized by many who have tested different compounds on their ability to target iAβ in AD treatment.

### 9.1. Bioactive Polyphenols

Notably, promising effects in iAβ clearance have been observed for naturally occurring, bioactive polyphenols, more specifically flavonoids, which have shown in numerous previous trials their antioxidant, anti-inflammatory, and autophagy-inducing properties and, therefore, promise to be of great potential in iAβ-targeting drug treatment [108,109,110,111,112,113,114,115,116,117,118,119,120]. The exact mechanisms through which polyphenols exert these neuroprotective capacities are not yet fully understood, though to some extent they can be led back to their aromatic phenolic groups, due to which they contain a varying number of functional hydroxyl groups that mediate their antioxidant effects by scavenging free radicals or by chelating metal ions [121]. At the same time, they possess the ability to cross the blood–brain barrier, which is a prerequisite for their pharmaceutical relevance [122]. The recognition of this potential has, amongst others, led to a trial in which 11 different natural polyphenol compounds were tested on their capability to reduce iAβ in genetically modified *Saccharomyces cerevisiae,* finding that a combination of baicalein and trans-chalcone significantly reduces iAβ_42_ and cellular ROS induced by Aβ42 in a synergistic manner [123]. Supporting this finding, it was reported that anthocyanin-enriched extracts from fruits of mulberry (*Morus alba* Linn.) ameliorate the cytotoxicity induced by iAβ oligomers in a transgenic mouse model [124]. Similarly, iAβ and oxidative stress were reduced in in vitro trials using modified polyphenol compounds, e.g., a nitroxide spin label linked to amyloidophilic fluorenes (spin-labeled fluorene, SLF) [125] or quercetin-modified gold–palladium nanoparticles [126]. Another compound shown to be effective is menadione sodium bisulfate (MSB), also known as vitamin K_3_, which is a less toxic analog of 1,4-naphthoquinone. MSB has similarly shown efficient inhibition of iAβ in HEK293T cells expressing the familiar AD Osaka mutation [127].

### 9.2. Cannabinoids

Another natural source for potential drug candidates in AD treatment is *Cannabis sativa* whose potential for therapeutic use has in recent years been shifted increasingly into the focus of research. *Cannabis* contains more than 100 kinds of cannabinoid compounds with the major classes being Δ9-trans-tetrahydrocannabidiol (THC) and cannabidiol (CBD). The neuroprotective effect of THC against iAβ accumulation has already been evidenced in several drug screening studies [43,128,129]. However, due to the psychoactive properties of THC, other non-psychoactive classes of cannabinoids, as well as artificial cannabinoids, have been tested as more suitable treatment options, and recent reports were able to demonstrate promising neuroprotective effects with an especially prominent effect on the reduction in iAβ [43,128,130]. Cannabinoids act as agonists on the cannabinoid receptors CB_1_ and CB_2_, which are part of the endocannabinoid system (ECS) that is involved in a number of important physiological processes such as memory and learning, brain plasticity, neuroinflammation, neuronal development, appetite regulation, etc. [131,132,133,134]. The ECS functions through retrograde neurotransmission, which means that post-synaptic neurons release endocannabinoids that bind (predominantly CB_1_ receptors) on the presynaptic neuron resulting in inhibited presynaptic calcium channel activation and subsequent presynaptic neurotransmitter release. The specific effect is then dependent on the neurotransmitter type and the respective cell type and part of the nervous system [135]. The exact mechanisms through which ECS is involved in neurotoxic processes, and in particular iAβ accumulation, still remain objects of research; therefore, its role and the effect of cannabinoids in neuroprotective mechanisms are also yet to be fully resolved.

### 9.3. Antibodies

Furthermore, several types of monoclonal antibodies have been developed that specifically target soluble oligomeric forms of Aβ, which have been found to be the more toxic species, compared with the actual amyloid plaques [136]. A strong candidate that even reached the clinical phase, but failed due to adverse effects, is bapineuzumab [137]. Based on the promising potential of bapineuzumab, researchers aimed at optimizing this approach by utilizing only the single-chain variable antibody fragment scFv-h3D6 derived from bapineuzumab, which is directed against the five N-terminal residues of Aβ, and, therefore, recognizes all the aggregation states of the peptide (monomers, oligomers, and fibrils) [138]. The effect of scFv-h3D6 on amyloid pathology was tested in 5-month-old 3xTg-AD female mice with the result that scFv-h3D6 treatment dramatically reduced iAβ amyloid pathology, resulting in preservation of cell density and amelioration of cognitive disabilities, which once more points to the significance of iAβ as a target for AD treatment. Interestingly, it was observed that the treatment was safe in terms of neuroinflammation and kidney and liver function, whereas some effects on the spleen were observed [139].

### 9.4. Other Treatments

The fact that iAβ is a strong drug target candidate has further been proven by the variety of substances that have been reported to reduce iAβ with accompanying neuroprotective effects. Interestingly, it was demonstrated that ablation of microglia by administration of colony stimulation factor 1 receptor (CSF1R) inhibitor PLX3397 led to a dramatic reduction in iAβ and neuritic plaque deposition in a mouse model, pointing to a causal relation between CSFR1 signaling and iAβ, which might reveal just another primary target in the focus of iAβ [140].

All of these various and recent approaches demonstrate the rising attention for iAβ as a target for the treatment of AD and have already proven a great potential for a range of potentially strong drug candidates. However, the presented selection of recent studies has been performed in vitro or in animal studies, and thus they need to be considered with some caution, as their effects on humans, adverse and beneficial, remain unexplored. However, even though we are still a long way from the actual application in humans, and in the past, many promising drugs failed the clinical test in the end, iAβ-targeting drugs have the potential to become the next big candidate in clinical AD treatment trials.

## 10. Future Perspectives

Since iAβ precedes the extracellular accumulation of Aβ [16,40,141], a new hypothesis had to be formulated, shifting from the classical “amyloid hypothesis” to the “intracellular Aβ hypothesis” [71]. As demonstrated, many studies revealed evidence for iAβ being a potent pathogenic agent. However, some results have to be addressed carefully, for example, regarding the proper identification of iAβ since antibodies such as the widely used 6E10 (targeting the N-terminus of Aβ) also recognize AβPP, also known as CTF99, the first product of the amyloidogenic APP cleavage. This cross-reactivity renders the isolated Aβ recognition difficult and gives ground for controversial discussion [142,143]. Future experiments need to unravel the relevance of iAβ exclusively, independent of extracellular amyloid β generation and aggregation. It urgently needs to be resolved if iAβ itself is sufficient to cause progressive neurodegeneration, and therefore, the use of complex models such as animal or human organoid models is required. In more detail, it needs to be understood, (1) which cells (which neuronal subtype) become iAβ-positive at early stages, (2) whether these cells have a higher risk of degeneration, (3) how neighbored cells react, (4) what the detailed molecular mechanisms in affected and neighbored cells are, and (5) what role inflammation plays in this context. Solidified determination of iAβ as an early pathogenic high-risk molecule with relevance for neurodegeneration will open up a new research field focusing on iAβ or early downstream pathways as a target for drug treatment approaches in order to fight AD and its tremendous symptoms.

## Figures and Tables

**Figure 1 ijms-23-04656-f001:**
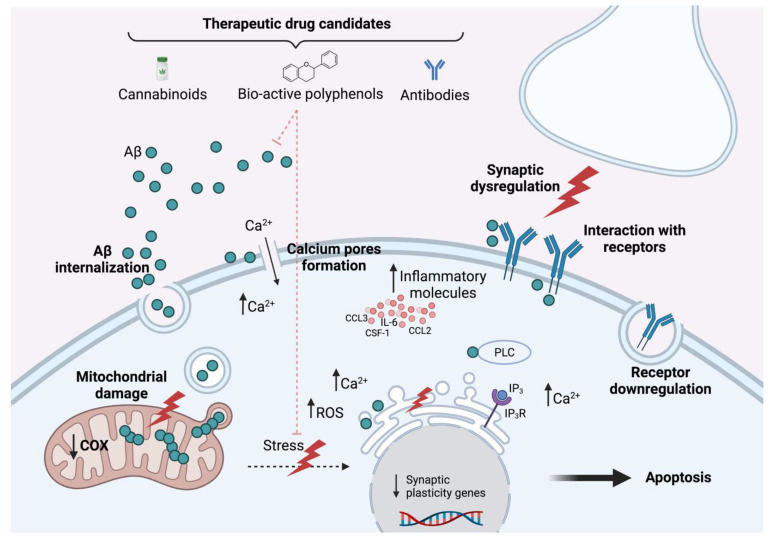
Uptake and effects of intracellular Aβ. While the precise mechanisms of intracellular Aβ accumulation remain unknown, recent reports indicate that the excessive accumulation results from internalization of extracellular Aβ via endocytosis or receptor interaction, but also mitochondrial uptake of long iAβ forms that are not secreted. Several studies have shown toxic effects on a number of cellular pathways, including calcium and synaptic dysregulation, inhibition of the ubiquitin–proteasome system, mitochondrial dysfunction, and activation of proinflammatory responses, resulting in cellular stress responses, and finally, apoptosis of the cell. (Created with BioRender.com).

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
