# Peer review of "Role of Intracellular Amyloid β as Pathway Modulator, Biomarker, and Therapy Target"

_ijms, 2022, doi:10.3390/ijms23094656_

Round 1

Reviewer 1 Report

On behalf of extracellular Aβ, a key player of the amyloid hypothesis that is stuck in treatment,
Intracellular Aβ is extremely interesting. This summary explains this point carefully.

In the section of application to the treatment, can the authors describe a little more about how each drug works on intracellular Aβ?

Author Response

On behalf of extracellular Aβ, a key player of the amyloid hypothesis that is stuck in treatment, Intracellular Aβ is extremely interesting. This summary explains this point carefully. In the section of application to the treatment, can the authors describe a little more about how each drug works on intracellular Aβ?

We thank you for your positive feedback and, with regard to your request, have revised the indicated part regarding the application in treatment. However, for most of the presented drug options, the mechanism how the respective compound works on iAβ is not fully understood. We have looked into publications to give more information regarding this where it is possible and indicated in the revised document, but unfortunately cannot give more description for most of the compounds as we do not want to include too much speculative and contradictive research results and a substantiated discussion would exceed the scope of this review.

Reviewer 2 Report

The current review presented by  Villarejo et al is inresting but certain quireis needs to be addressed before acceptence. I would recommend minor revision. 

  1. The statment made by the reviewers does not backed by authors with proper citations for e.g page 1 line 40- 41, According to the amyloid hypothesis, amyloidogenic plaque formation is caused by 40
    the dysregulation of the Aβ peptide homeostasis due to altered processing of the amyloid 41 precursor protein (APP). 
  2. It would be great to cover the role of prefibrillar Aβ species in AD.
  3. Authors covers the role of polyphenols, canabinnoids and antibodeis to target Aβ as a therapeutics for AD. It would be intesting to cover the role of vitamins and osmolytes to traget Aβ (PMID: 4882616, PMID: 30738899).
  4. It would be great to incldue a section on role of Aβ in regaulating immuno response and various markers based on immunology assays. 

Author Response

The current review presented by Villarejo et al is inresting but certain quireis needs to be addressed before acceptence. I would recommend minor revision.

Thank you for your review and your helpful feedback, we appreciate it a lot and, in the following, answer point-by-point to each aspect.

  1. The statment made by the reviewers does not backed by authors with proper citations for e.g page 1 line 40-41, According to the amyloid hypothesis, amyloidogenic plaque formation is caused by 40 the dysregulation of the Aβ peptide homeostasis due to altered processing of the amyloid 41 precursor protein (APP).

As suggested, literature references supporting the statements have now been added to the manuscript and the text has been modified accordingly.

  1. It would be great to cover the role of prefibrillar Aβ species in AD.

Information regarding the role of prefibrillar Aβ species has been added to the manuscript in the introduction chapter.

  1. 3. Authors covers the role of polyphenols, canabinnoids and antibodeis to target Aβ as a therapeutics for AD. It would be in testing to cover the role of vitamins and osmolytes to traget Ac (PMID: 4882616, PMID: 30738899).

We sincerely thank you for this suggestion as it is indeed a very interesting aspect. Unfortunately, the first PMID is leading to a publication that is not related to the topic and probably not the intended number. Further, while being highly interesting, the publication of the second PMID does not focus on intracellular Aβ in particular. In general, we were not able to find sufficiently supportive data that focusses specifically on intracellular Aβ to include a whole new chapter. With regard to the fact that our review should give a short overview of the latest progress and the currently strongest discussed compounds, we hope that our decision to omit this new chapter is accepted. Otherwise, we are happy to receive more input on this topic.

  1. It would be great to incldue a section on role of Aβ in regaulating immuno response and various markers based on immunology assays.

We appreciate your suggestion and we have included additional studies addressing the role of the immune system in the chapter “Intracellular Aβ as driver of neuroinflammation“.
